# Iron disproportionation in peridotite fragments from the mantle transition zone

Fabin Pan[1], Xiang Wu[1] ✉, Chao Wang [1], Yanfei Zhang[1], Yiping Yang[2], Xiaobo He [3], Chong Jin[4], Lian Zhou[1], Hongfei Zhang[1], Hongping He [2] & Junfeng Zhang [1] ✉

Previous high-temperature-pressure experiments predicted metallic iron's potential presence in the deep mantle below 250 km, arising from ferrous disproportionation in silicates, which could profoundly impact the redox environment and physicochemical properties. However, direct natural petrological evidence has been lacking, except scant clues like Fe-alloy inclusions in ultradeep diamonds. Here we present peridotite fragments, found in Cenozoic basalts from eastern China, containing decomposed Na-rich majoritic garnets (from depths of 410-550 km) and olivine with $Fe^0$-spinel-bearing inclusions, likely originated from retrograded wadsleyite/ringwoodite. Enriched Zn-Sr isotopic compositions of the decomposed garnet indicate an origin associated with the stagnant Pacific slab in the mantle transition zone. Disproportionation of iron is evidenced by widely distributed submicron-sized spherical Fe-Ni alloys and $Fe^{3+}$-rich ($Fe^{3+}/\Sigma Fe = 0.35$-$0.40$) olivine. These findings provide compelling evidence for recycling of stagnant slab components in the eastern Asia big mantle wedge (BMW), and iron disproportionation in the deep mantle.

Iron is the most abundant redox-sensitive element in the deep mantle, and the relative concentrations of $Fe^{3+}$, $Fe^{2+}$, and $Fe^0$ can well record the variations in oxygen fugacity[1]. These variations play a crucial role in the recycling of volatile elements like carbon, hydrogen, and sulfur between Earth's interior and surface[2,3]. A prevailing notion is that considerable portions of the mantle below 250 km are saturated with an iron–nickel rich metal phase formed by iron disproportionation in silicates, with the redox state controlled by the iron–wüstite (IW) buffer[4-6], as oxygen fugacity decreases with depth in Earth's interior[7]. This pressure-driven autoredox reaction is evidenced in samples from both Earth and Moon's surfaces, particularly in meteorite impact zones[8,9]. Some ultradeep diamonds (>300 km) contain native iron-alloy inclusions attributed to this deep mantle process[10]. Although deep upper mantle and uppermost lower mantle silicate minerals often exhibit high ferric iron content ($Fe^{3+}/\Sigma Fe = 0.15$-$0.87$)[11-13], native iron-bearing phases have not been concurrently found. This discrepancy is likely due to oxidation caused by infiltration of oxidized carbonated fluids/melts, rather than disproportionation of iron in a reducing setting[11]. Thus, the underlying mechanism governing iron's redox state variation in the deep mantle remains enigmatic, lacking direct petrological evidence.

Here, we present five "garnet" lherzolite xenoliths extracted from Cenozoic alkaline basalts located in Zhejiang Province, Southeast China (Supplementary Fig. 1). In this area, Oligocene (27–23 Ma) basanite-tephrite rocks are distributed in the interior of the continent, while Miocene to Pliocene (17–2 Ma) basanite, alkaline basalt, and olivine tholeiite are found along the coast. The eruption ages of these rocks decrease oceanward, with an -100-km spatial gap, correlating temporally and spatially with repeated retreats and rollbacks of the westward subducting Pacific plate during the Tertiary period[14]. The host rock for the studied xenoliths is nephelinite, with $SiO_2$ ranging from 39.99 to 40.76 wt.%, $Al_2O_3$ from 9.55 to 10.30 wt.%, MgO from

[1]State Key Laboratory of Geological Processes and Mineral Resources, and School of Earth Science, China University of Geosciences, Wuhan, China. [2]State Key Laboratory of Deep Earth Processes and Resources, Guangzhou Institute of Geochemistry, Chinese Academy of Sciences, Guangzhou, China. [3]Marine Science and Technology College, Zhejiang Ocean University, Zhoushan, China. [4]Zhejiang Institute of Geosciences, Hangzhou, China. ✉ e-mail: wuxiang@cug.edu.cn; jfzhang@cug.edu.cn

12.12 to 14.17 wt.% (Mg# = 64−67), CaO from 10.17 to 10.76 wt.%, $Na_2O$ from 4.39 to 5.02 wt.%, and a $Na_2O/K_2O$ ratio of 2.3 to 2.5 (Supplementary Data 1). The ferric iron contents ($Fe^{3+}/\Sigma Fe$) of the nephelinite range from 0.25 to 0.31, which is consistent with the Cenozoic highly oxidized intraplate basalts in eastern China[15]. The contemporaneous nephelinite in eastern China have been suggested to originate from the mantle transition zone (MTZ)[16]. The "garnet" lherzolite xenoliths (Fig. 1a) exhibit minimal hydrothermal alteration, appearing fresh with a porphyroclastic texture. They consist mainly olivine ($Ol_1$, ~45 vol.%), orthopyroxene (~25 vol.%), clinopyroxene (~20 vol.%), and kelyphitized garnet (K-Grt, ~10 vol.%).

## Results and discussion
### Na-rich majoritic garnet derived from the MTZ
The "garnets" in the lherzolites are completely kelyphitized, displaying a distinct core-rim structure (Fig. 1a). The rim is composed of anhedral orthopyroxene, clinopyroxene, spinel, and plagioclase (Fig. 1b and Supplementary Fig. 2), with variable, out of equilibrium chemical compositions (Supplementary Data 2). These minerals result from rapid exhumation via the reaction between garnet and olivine (Reaction 1: Grt + $Ol_1$ = Opx + Cpx + Sp/Pl), with plagioclase representing the terminal phase of this retrograde reaction. Notably, the presence of plagioclase suggests sodium and potassium loss from the K-Grt core (Supplementary Fig. 2).

The K-Grt core dominantly consists of orthopyroxene, olivine ($Ol_2$), spinel, Fe-Ni alloys, and a Na-rich matrix (Fig. 1c-d and Supplementary Fig. 3). Fe-Ni alloys are widespread in the core but tend to diminish near the rim (Fig. 1b). Two domains are discernible in the core: Opx + Sp + Na-rich matrix and $Ol_2$ + Sp + Na-rich matrix. The Na-rich matrix in the decomposed garnet is dominantly composed of plagioclase crystallites with variable $Na_2O$, $K_2O$, and CaO contents (Supplementary Fig. 4). Complex multiphase symplectite-bearing coronae around breakdown garnets are often found in exhumed mantle xenoliths formed through rapid decompression reaction[17,18]. Previous studies on garnets from Zinst have interpreted sodium enrichment in the kelyphite garnet as a result of partial metasomatism by external Na-rich carbonate-bearing melts or fluids in the uppermost mantle (<60 km)[17]. However, this is inconsistent with the formation of the K-Grt core in this study, despite the similar microstructure to the isochemical breakdown of garnet observed in those studies. All K-Grt cores in this study contain visible Fe-Ni spherules, indicating highly reduced conditions, which contrasts with the highly oxidized host nephelinite and precludes a reaction mechanism involving external Na-rich carbonate-bearing melts or fluids. Instead, the K-Grt core is likely the breakdown product of the parental garnet itself (Reaction 2: Grt = Opx + Sp + An ± Cpx)[18]. In addition, the host nephelinite shows significant enrichment in lithophile elements, consistent with the Cenozoic OIB-type intraplate basalts in eastern China (Supplementary Fig. 5a). However, in-situ trace element analyses of the core show typical lithospheric garnet features, i.e., very low lithophile element concentrations and enrichment of heavy rare earth elements (Supplementary Fig. 5 and Supplementary Data 3), indicating a relatively closed isostoichiometric system and minimal exchange with external fluids/melts. Similar isochemical breakdown of natural garnet has been previously observed in orogenic garnet peridotites[19]. The widespread occurrence of olivine and the Na-rich matrix in the decomposed garnet likely formed by decompression-induced incongruent melting[18,20].

Seven K-Grt cores from three xenoliths were selected for major element analysis of the parental garnet. To mitigate the challenge of inhomogeneous composition caused by kelyphitization, a 25-μm diameter electronic beam was used for electron microprobe measurement, taking the average of more than fifteen different areas in each K-Grt core as the representative composition of the parental garnet (Supplementary Data 4). The parental garnets show compositions of $SiO_2$ (44.42−45.00 wt.%), $Al_2O_3$ (22.33−23.05 wt.%), MgO (18.62−20.41 wt.%), $Cr_2O_3$ (1.60−1.78 wt.%), CaO (4.84−5.69 wt.%), FeO (3.39−3.90 wt.%), $TiO_2$ (0.18-0.24 wt.%), along with variable $Na_2O$ (0.75−2.41 wt.%) and $K_2O$ (0.11−0.73 wt.%) contents. The variations in sodium and potassium contents are attributed to partial migration into the K-Grt rim (Fig. 1b and Supplementary Fig. 2). Standard formula calculation shows high Si+Ti (3.13−3.15 per formula unit (p.f.u.)), along with low Mg+Fe+Ca+Mn+Na+K (2.83−2.96 p.f.u.) contents in the

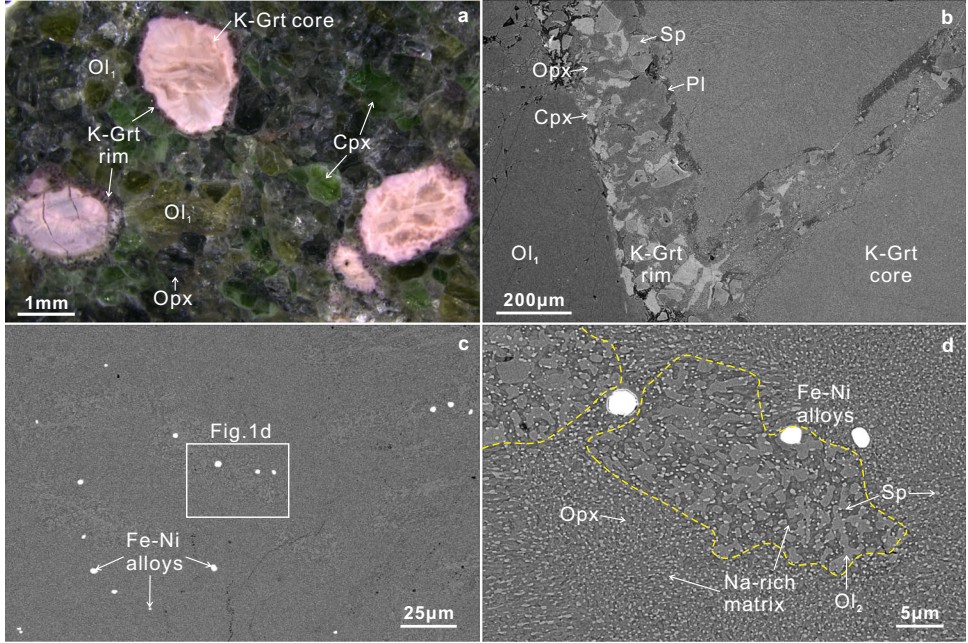

**Fig. 1 | Backscattered electron micrographs illustrating the kelyphitized garnet. a** "Garnet" lherzolite consists of kelyphitized garnet (K-Grt), olivine ($Ol_1$), clinopyroxene (Cpx), and orthopyroxene (Opx). **b** The K-Grt rim consists of Opx, Cpx, spinel (Sp), and plagioclase (Pl), lied between $Ol_1$ in the host lherzolite and the K-Grt core. **c**, **d** The K-Grt core exhibits a nanometer-sized domain composed of Opx + Sp + Na-rich matrix, juxtaposed with domains composed of $Ol_2$ + Sp + Na-rich matrix featuring slightly larger grains (enclosed within the yellow dashed lines). Spherical Fe-Ni alloys are widely distributed in both domains.

parental garnets. Given that a substantial portion (10-30%) of iron in majoritic garnets[11] should exist as $Fe^{3+}$, this imbalance would be exacerbated under the conditions of the deep mantle. This suggests a loss of highly mobile cations, likely Na and K, during kelyphitization. Evidence for this loss of Na and K from the K-Grt core is apparent in the elemental maps of the K-Grt rim (Supplementary Fig. 2). As these components were not included in the garnet reconstruction process, the proportions of Na-rich majorite represent the lowest conceivable contents for the studied garnets.

The formation of majorite (Maj) at high pressure relies on two substitution mechanisms[21]: Mg-Maj substitution in a peridotitic composition (Reaction 3: $2Al^{3+} = M^{2+} + Si^{4+}$) and Na-Maj substitution in an eclogitic composition (Reaction 4: $Al^{3+} + M^{2+} = X^+ + Si^{4+}$), where $M^{2+}$ is for $Mg^{2+}$, $Fe^{2+}$, $Ca^{2+}$, $Mn^{2+}$ and $X^+$ is for $Na^+$ and $K^+$, respectively. High-temperature and high-pressure experiments have demonstrated that the composition of majoritic garnet is pressure-dependent[13,21–23], making it a reliable and widely accepted pressure indicator for deep mantle materials (at depths of 300–660 km). The parental garnet reconstructed in this study belongs to the group of eclogite-derived majoritic garnet (Supplementary Fig. 6a, b). Phase variations of synthetic eclogite were performed in a simplified diopside (Di)–hedenbergite (Hd)–jadeite (Jd) system over a pressure range of 7–24 GPa, yielding a linear correlation between the Na content in majoritic garnet and pressure ($P_{B08}$)[23]. Using this empirical barometer, we obtained an equilibrium pressure of 14.5-19.0 GPa (Supplementary Data 4). Additionally, a new barometer ($P_{B17}$) based on high-temperature and high-pressure experiments of basaltic and pyroxenitic compositions[21], yielded a comparable equilibrium pressure of 13.3–18.9 GPa for the parental majoritic garnet (Fig. 2a). Therefore, the high Na-K contents of the parental majoritic garnet imply an origin in the MTZ.

Seismic tomography shows the subducting Pacific slab is stagnant in the MTZ beneath eastern Asia[24–26]. The Na-rich majoritic garnet lherzolite in this study is hosted within Cenozoic basaltic volcanics located above the stagnant slab. Previous studies on the isotopic compositions of the basalts have speculated on the possible recycling of materials from the stagnant slab[14,27]. To trace the origin of the decomposed garnets, we analyzed four powdered samples (300 mg for each) for whole-rock Sr, Nd, and Zn isotopic compositions, given

the notably low Sr (<5 ppm) and Nd (<1 ppm) concentrations (Supplementary Data 3). The results (Supplementary Data 5) reveal depleted Nd isotopic compositions ($^{143}Nd/^{144}Nd = 0.5131-0.5132$) and slightly enriched Sr isotopic compositions ($^{87}Sr/^{86}Sr = 0.7039-0.7042$). Besides, these decomposed garnets show elevated zinc isotopic compositions ($\delta^{66}Zn = 0.32-0.33‰$), compared to those ($\delta^{66}Zn = 0.2-0.3‰$) of the mantle and oceanic crust[28,29]. Their zinc isotopic compositions are consistent with those of the Cenozoic intraplate basalts in eastern China, which are proposed to have significant carbonated eclogite (subducted Pacific MORB residuum) contributions in their mantle source[28,29]. The Sr and Nd isotopic feature differs from the typical Sr-Nd isotopic evolution of the depleted MORB mantle and the enriched mantles influenced by subducted continental crust (Fig. 2b). Instead, the Sr−Nd−Zn isotopic characteristics of these majoritic garnets are similar to those of the carbonated MORB (Fig. 2b and Supplementary Fig. 6c), suggesting that they likely form via reactions with components from the stagnant Pacific slab within the MTZ[24–26]. Interaction with peridotitic components during these reactions may have led to enrichment in Mg and Cr, and depletion in Ca and Ti, along with elevated Na and K contents, resulting in the observed transitional geochemical features[21].

## Olivine originated from the MTZ

A third type of olivine ($Ol_3$) is found intergrown with relatively large kelyphitized garnets (Fig. 3a and Supplementary Fig. 7). These olivine inclusions are black, ellipsoidal or irregular in shape, ranging in size from 0.5 to 3.0 mm. Backscattered electron micrographs show widespread tiny Fe-Ni alloy inclusions within these olivine grains (Fig. 3b). High-angle annular dark-field scanning transmission electron micrographs also reveal many elliptic inclusions in olivine-$Ol_3$, ranging from nearly 10 to 100 nm (Fig. 3c and Supplementary Figs. 8a–f). Additionally, one octahedral inclusion (a typical shape for spinel) was discovered coexisting with an elliptical inclusion and two Fe-Ni alloy inclusions in the same focus ion beam (FIB) foil (Fig. 3c and Supplementary Fig. 8g–l). Their compositions are detailed in Supplementary Data 6. High-resolution transmission electron microscope (HRTEM) images and corresponding fast Fourier transform (FFT)\nano-beam electron diffraction (NBD) reveal comparable Mioré Patterns between the octahedral inclusion and elliptic inclusions (Supplementary

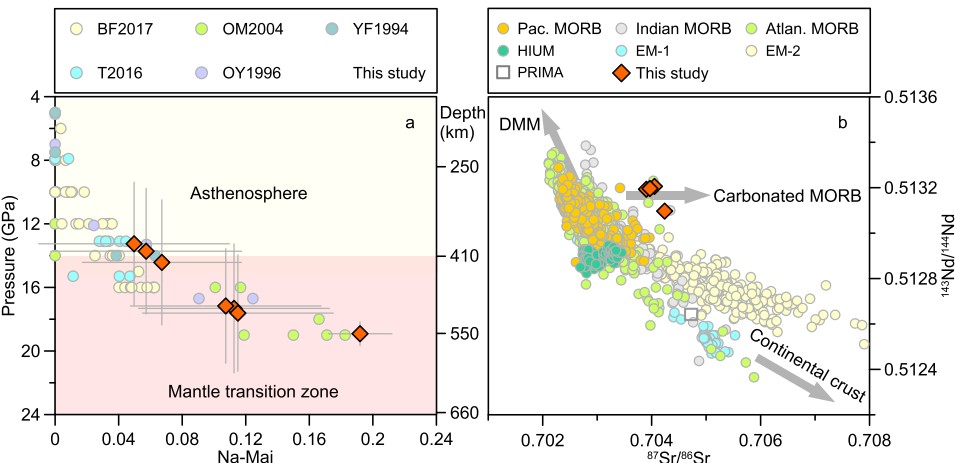

**Fig. 2 | Na−Maj contents and Sr−Nd isotopic compositions of the kelyphitized garnet cores. a** Na-Maj content plotted against experimental pressure on basalts. The experimental data were sourced from Beyer and Frost (2017)[21], Thompson et al.[42], Okamato and Maruyama[54], Ono and Yasuda[22], and Yasuda and Fujii[55]. The pressure of the K-Grt core is calculated using the method proposed by Beyer and Frost (2017), with a 1σ uncertainty of -0.27 GPa[21]. The 2σ uncertainties for both the measured Na-Maj contents and the calculated pressures are provided in

Supplementary Data 4. **b** Sr and Nd isotopic compositions of the kelyphitized garnet cores and typical basalts derived from the Earth's convective mantle. The summarized Sr and Nd isotopic data for Pacific, Indian, and Atlantic MORBs, as well as HIUM, EM-1, and EM-2, are sourced from Geochemical Rock Database (https://doi.org/10.25625/0SVW6S). The primitive mantle (PRIMA) value is taken from Hofmann[56].

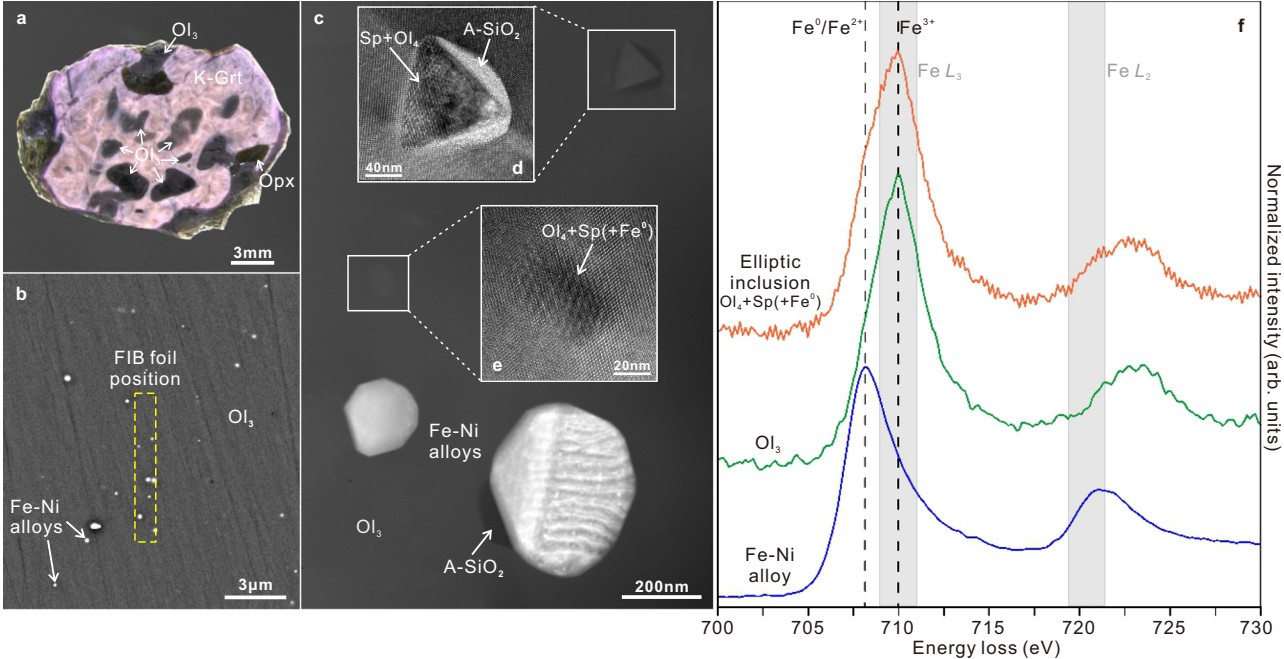

**Fig. 3 | Fe–Ni alloy and spinel-bearing inclusions in olivine ($Ol_3$) and the valence state of iron in them. a** Photograph of a peridotite fragment depicting the intergrown of olivine ($Ol_3$) and a K-Grt. **b** Backscattered electron micrograph showing ubiquitous tiny Fe-Ni alloy inclusions in $Ol_3$, with the dotted square indicating the cutting position of a foil using the focus ion beam (FIB). **c** High-angle annular dark-field scanning transmission electron micrograph (HAADF-STEM) of the foil in **b** showing two Fe-Ni alloy inclusions, an elliptic inclusion, and an octahedral inclusion in $Ol_3$. Amorphous silicon dioxide ($A-SiO_2$) is observed next to the octahedral inclusion and an Fe-Ni alloy inclusion. The insets **d** and **e** in **c** are HRTEM micrographs of the octahedral inclusion and the elliptic inclusion. **f** Representative EELS of Fe $L_2$, $L_3$ core loss edge of a Fe-Ni alloy inclusion, an elliptic inclusion, and $Ol_3$. The peak position of Fe $L_3$ at 710 eV indicates trivalent iron ($Fe^{3+}$), while at 708.4 eV, it suggests divalent iron and native ion ($Fe^0/Fe^{2+}$)[57], respectively.

Fig. 8m–o). Their NBD patterns can be indexed as a mixture of olivine ($Ol_4$) and spinel. Amorphous silicon dioxide ($A-SiO_2$) is found adjacent to both the octahedral inclusion and an Fe-Ni alloy inclusion (Fig. 3c).

The valence states of iron in olivine ($Ol_3$), elliptic inclusion and Fe-Ni alloy inclusion were measured using electron energy loss spectroscopy (EELS). $Fe^{3+}$ content was assessed using a modified integral intensity ratio of the Fe $L_2$ and $L_3$ white lines, yielding $Fe^{3+}/\Sigma Fe$ ratios of 0.35 for olivine ($Ol_3$) and 0.40 for the elliptic inclusion (Fig. 3d). Molecular formula calculations indicate total cations of the olivine ($Ol_3$) and elliptic inclusions within the range of 3.10-3.29 (Supplementary Data 6), higher than those of standard olivine and spinel ($A_2BO_4$ with total cations of 3), implying the likely presence of minor nano-Fe-Ni alloys. These nano-$Fe^0$ dots were sometimes observed in HRTEM images. In addition, the elliptic inclusion presents a more pronounced shoulder peak at 708.4 eV compared to the olivine ($Ol_3$) (Fig. 3d), demonstrating the presence of more nano-$Fe^0$. Analysis of the relative contents of Si, Al, and Cr indicates spinel proportions ranging from 7% to 27% in the elliptic inclusions and nearly 20% olivine ($Ol_4$) in the octahedral inclusion (Supplementary Data 6).

High-temperature-pressure experiments in natural pyrolite show that the high-pressure polyform olivine (HP-Ol) (i.e. wadsleyite/ringwoodite), dominant in the MTZ, often contains significant aluminum (up to 1 wt.%) in their modified-spinel/spinel crystal structures for a high-pressure $AB_2O_4$ stoichiometry[30,31]. In the upwelling asthenosphere, these aluminum-rich HP-Ol minerals become unstable and tend to decompose into olivine (($(Mg,Fe)_2SiO_4$) and spinel (($(Mg,Fe)Al_2O_4$). Thus, both the elliptic inclusions ($Ol_4$ + Sp) and the octahedral inclusion (Sp + $Ol_4$) with comparable Mioré Patterns in the studied olivine ($Ol_3$) are attributed to be the residual phases of retrograded HP-Ol under descending pressure. Totally, as the Na-rich majoritic garnet host originated from depth of 410–550 km, all the olivine ($Ol_3$) crystals were considered to be the final phase of retrograded HP-Ol originally captured in the MTZ.

## Iron disproportionation in deep mantle silicates

Super-reduced mineral assemblages, such as native elements, Fe-carbides, and silicides, have been documented worldwide in various upper mantle lithologies, often formed by reactions between reducing C–H–O fluids and surrounding silicates[32–35]. The reaction between olivine and highly reducing C–H–O fluids releases significant iron, leading to the high Mg# (up to 99) (Mg# = 100 × Mg/(Mg+Fe)) in residual olivine/wadsleyite/ringwoodite[36]. However, contrary to this trend, olivine-$Ol_3$ and elliptic inclusions show relatively low Mg# (89-92). The release of $CO_2$ and $H_2O$ fluids during these redox reactions would typically lower the melting temperature and induce partial melting of surrounding silicate materials. Nevertheless, the high alkali contents of decomposed garnet, low Mg# olivine, and fertile clinopyroxene of the "garnet" lherzolite xenoliths, preclude widespread partial melting. Additionally, Fe–Ni spherules are exclusively found in olivine inclusions ($Ol_3$) and decomposed majoritic garnets, absent in the host lherzolite, indicating a distinct formation mechanism from typical redox reactions between reducing fluids and surrounding silicates.

$Fe^{3+}/\Sigma Fe$ in olivine is typically low (<0.01) in the shallow upper mantle[37,38]. However, under high pressure and oxidizing environments, the presence of water in olivine can elevate concentrations of $Fe^{3+}$ ($Fe^{3+}/\Sigma Fe$ up to 0.15)[37,38]. This ratio can escalate further in wadsleyite, reaching up to 0.44 in high-pressure hydrous experimental systems[37]. High-pressure and high-temperature experiments suggest that iron disproportionation reaction can lead to significant ferric iron content in wadsleyite and ringwoodite, enhanced by water dissolution and Al incorporation[4,37,39]. The $Fe^{3+}/\Sigma Fe$ values in this study resemble those observed in hydrous experimental wadsleyite under the MTZ conditions[37]. Thus, the coexistence of $Fe^0$, $Fe^{2+}$, and $Fe^{3+}$ in olivine

inclusions ($Ol_3$) and elliptic inclusions confirms the formation of metallic and ferric iron via iron disproportionation in the deep mantle. Presence of amorphous silicon dioxide next to the octahedral inclusion and Fe-Ni alloy inclusion (Fig. 3c) is likely the byproduct of the disproportionation reaction: $3Fe_2SiO_4 = 2(Fe^{3+}, vacancy) SiO_4 + 4Fe^0 + SiO_2 + O_2$ (Reaction 5).

Decomposed majoritic garnets show clearly low Fe contents (0.20–0.23 p.f.u.) and high Mg# of 90–91, contrasting with Na-rich majorite inclusions (Fe: 0.6–1.1 p.f.u., Mg#: 30–70) in ultradeep diamonds[40]. The significant lower Fe contents in observed decomposed majoritic garnets relative to other eclogite-derived majorites suggest Fe release through iron disproportionation, with $Fe^{3+}$ retained while $Fe^0$ exsolves as metallic phase. These Fe–Ni alloys are believed to remain as small, isolated intergranular grains within their host in the upper mantle[41]. This hypothesis is supported by the presence of native Fe–Ni alloy inclusions in both the olivine inclusions ($Ol_3$) and decomposed majoritic garnets observed in this study. These metallic Fe-Ni alloys have high stability in the deep mantle characterized by a low oxygen fugacity environment. Our observations show that they tend to aggregate and persist within their hosts, rather than undergoing neutralization reactions and disappearing with ferric iron during the retrogression of wadsleyite/ringwoodite and majoritic garnet in the upwelling asthenosphere. These petrological features strongly support the widespread of metallic Fe-Ni phases and iron disproportionation in the deep mantle.

### Implications for recycling from the stagnant Pacific slab

Evidence from previous experiments and ultradeep diamond inclusions indicates that carbonatite melts associated with a subducting slab can form in the deep upper mantle and subsequently subject to redox freezing in ambient peridotite[2,3,42]. The upwelling of metasomatized mantle initiates partial melt extraction around 250–300 km[2,3,43], leading to chemical and isotopic heterogeneity in the upper mantle. These processes emphasize the vital role of redox melting in the recycling of subducted slabs. Here we propose a schematic model to illustrate the recycling of Na-rich majoritic garnets in the big mantle wedge of eastern Asia[25,26] (Fig. 4). The envisioned recycling process is facilitated by small-scale upwellings ('return flow'[44,45]) originating from the MTZ in response to the arrival of the subducting Pacific slab. In the subducting oceanic crust, one major driving force for this process is the increased negative buoyancy in response to the mafic rocks-eclogite transformation in the crust[46]. As

depth increases, the volume fraction of the garnet component rises due to the dissolution clinopyroxene into garnet, eventually forming a monomineralic majoritic garnetite crust at MTZ depths (pressure > 14 GPa)[47]. This monomineralic majoritic garnetite could be captured by surrounding peridotite composed of Mg-Majorite and retrograded high-pressure polymorph of wadsleyite/ringwoodite (RHP-Ol) in the MTZ. The upwellings initially carry majoritic garnets up to nearly 300 km depth, where they are inherited by the asthenosphere lherzolite. Subsequently, the eruption of basaltic volcanics, derived from the asthenosphere lherzolite, rapidly transport these materials to the surface. Slab subduction and asthenosphere upwelling drive vertical mantle convection, promoting mechanical and chemical mixing in the mantle.

Oceanic slab subduction has traditionally been seen as a mechanism for introducing oxidized crustal materials back into the Earth's interior, leading to significant oxidization in the deep mantle[48–50]. However, the vigorous disproportionation of ferrous iron to native iron and ferric iron preserved in the studied Na-rich majoritic garnet and RHP-Ol indicates that the redox state of the MTZ remains highly reduced. The Fe and Ni contents in the Fe-Ni alloys were measured by STEM-EDS (Supplementary Data 6). The Fe/Ni ratios of the Fe-Ni alloys in Na-rich majoritic garnets range from 247–280 in the high-Na grains to 20–25 in the low-Na grains, indicating progressively increasing oxygen fugacity relative to the ΔIW buffer[7]. In contrast, the Fe-Ni alloys in the RHP-Ol are enriched in nickel, with Fe/Ni ratios of 0.44-0.53 (Supplementary Data 6). This enrichment of metallic Ni in Fe–Ni phases within the RHP-Ol indicates that the oxygen fugacity in their originating MTZ is only slightly below the IW buffer[7]. The ferric iron content ($Fe^{3+}/\Sigma Fe$ = 0.35–0.40) in the RHP-Ol is comparable to reported values in wadsleyite[37], implying that disproportionation of iron in the MTZ is probably independent of the arrival of subducting materials. Our discovery provides compelling evidence supporting the widespread presence of metallic Fe-Ni phases and iron disproportionation in the deep mantle.

## Methods
### Electron microscope observation
The back-scatter electron (BSE) images were obtained using a FEI Apero S scanning electron microscope at the Stare Key Laboratory of Geological Processes and Mineral Resources (GPMR) in China University of Geosciences (Wuhan). The images were acquired with an accelerating voltage of 10 kV, a spot size of 10, an emission current of

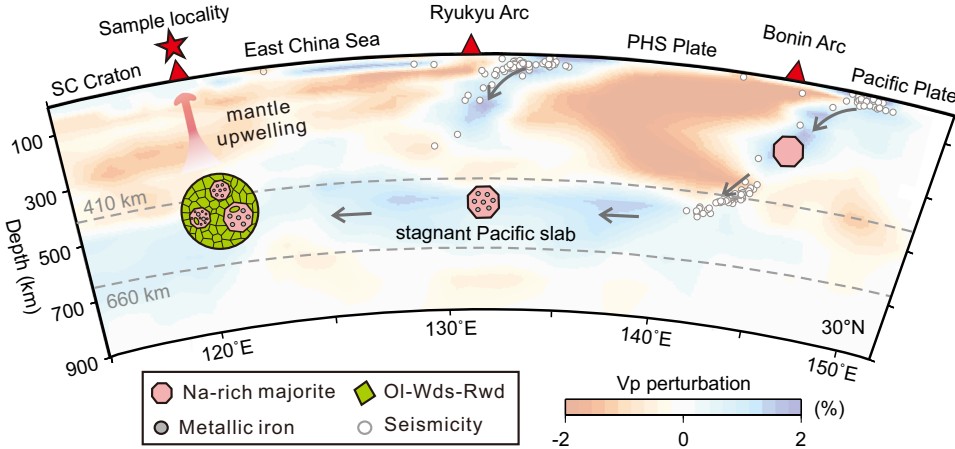

**Fig. 4 | Schematic model illustrating the recycling of peridotite fragments from the MTZ of the eastern Asia BMW.** Na-rich majoritic garnets, originating from the subducting Pacific slab, had metallic iron inclusions formed by iron disproportionation under low oxygen fugacity conditions during the stagnation of the Pacific slab in the MTZ. Subsequently, these garnets were entrained by ascending surrounding mantle predominantly composed of wadsleyite/Ringwoodite (Wds/Rwd), and transported to the surface of South China Craton as peridotite fragments by Cenozoic basaltic magma. The P-wave seismic image is adapted from Wei et al.[25]. SC craton: South China craton; PHS plate: Philippine Sea plate.

about 135 μA, and a working distance of 10 mm. The transmission electron microscopy (TEM) specimens were prepared utilizing the Focus Ion Beam (FIB) system (Helios G4 CX, ThermoFisher Scientific) equipped with a Transmission Kikuchi Diffraction (TKD) system at GPMR. Initially, a predefined area of approximately 20 μm² was coated with platinum (Pt), followed by precise cutting of its surroundings to a depth of around 10 μm using a gallium (Ga) ion beam. Subsequently, the resulting foil was delicately retrieved using EasyLift, an in-situ tungsten (W) probe inside the FIB, and mounted onto a TEM copper (Cu) grid (Omniprobe, Oxford Instrument). The extracted samples were thinned to 150 nm using a Ga ion beam at 30 kV, with beam currents ranging from 9.3 nA to 80 pA during processing, followed by additional steps at 5 kV with a beam current of 43 pA and 2 kV with a beam current of 23 pA for final processing. The analyses of the K-Grt cores involved using the TKD map dataset, generating band contrast (BC) images, phase maps, Euler images, and elemental maps.

## Major and trace elements analyses

Whole-rock nephelinite samples were analyzed for major elements, using conventional X-ray Fluorescence (XRF) method at the State Key Laboratory of Geological Processes and Mineral Resources (GPMR), China University of Geosciences, Wuhan. Loss on ignition (LOI) was determined by measuring the weight loss after drying the samples at 1000 °C. The analytical uncertainty for major element analysis is generally <5%. The ferric iron contents ($Fe^{3+}/\Sigma Fe$) in the nephelinite were measured using wet chemistry techniques. Trace elements, including rare earth elements (REE), were measured using an Agilent 7500a inductively coupled plasma mass spectrometer (ICP-MS) at GPMR. The relative error for most trace elements is below 5%, with the exception of Cr, Ni, Pb, and Zn, which have a relative error of approximately 10%.

Major element compositions of minerals and elemental mapping of the K-Grt rims were determined at GPMR using a JEOL JXA-8230 Electron Probe Micro-Analyzer (EPMA) equipped with five wavelength-dispersive spectrometers (WDS). Prior to analysis, the samples were coated with a thin conductive carbon film. Typically, an accelerating voltage of 15 kV, a beam current of 20 nA, and a spot size of 1–5 μm were used for analyzing minerals in the host peridotites and the K-Grt rims. Data were corrected online using a ZAF (atomic number, absorption, fluorescence) correction procedure to ensure accuracy. The peak counting time for elements such as Na, Mg, Al, Si, K, Ca, Fe, P, and Cr was set at 10 s, while for Mn, Ti, it was extended to 20 s. Background counting time was maintained at one-half of the peak counting time on both high- and low-energy background positions. The following standards were used for element-specific calibrations: Jadeite (Na), Almandine (Si, Mg, Fe in K-Grt cores), Olivine (Si, Mg, Fe in other minerals), Corundum (Al), Diopside (Ca), Sanidine (K), Rutile (Ti), Rhodonite (Mn), Apatite (P), Chromium Oxide (Cr), Pentlandite (Ni). The detection limit for major elements is generally less than 160 ppm, while the analytical uncertainty of the total amount of major elements is generally better than 2 wt.%.

Seven K-Grt cores, denoted M1-M7, were selected for major and trace element analyses in this study. To mitigate the challenge of inhomogeneous composition caused by kelyphitization, a meticulous approach was employed. Electron microprobe measurements were conducted using a substantial (25-μm) diameter electronic beam, averaging data from over fifteen different areas in each K-Grt core to represent the composition of the parental garnet. It is worth noting that all other measuring conditions remained consistent with those applied to the K-Grt rims. Analysis of the K-Grt cores revealed a depletion in Mg+Fe+Ca+Mn+Na+K content, ranging from 2.83 to 2.96 p.f.u., and an excessive Si+Ti content, ranging from 3.13 to 3.15 p.f.u., according to the garnet/majorite formula.

The trace element compositions of the K-Grt cores were analyzed on polished thick sections using LA-ICP-MS (GeoLas 2005+ Agilent

7500a) at GPMR. Laser sampling was performed using a spot size of 44 μm and a laser energy of 60 mJ. Each analysis included approximately 20–30 s of background acquisition (from a gas blank), followed by 50 s of data acquisition from the sample. The element contents were calibrated against USGS reference glasses (BCR-2G, BHVO-2G and BIR-1G), and normalization of the sum of all metal oxides to 100 wt% was applied. Off-line selection and integration of background and analyzed signals with time-drift correction and quantitative calibration were performed using ICPMSDataCal[51]. The smooth signal intensity observed during the analysis of the K-Grt cores indicates no significant contamination by inclusions or other mineral phases. Detailed data from major and trace element analyses are listed in Supplementary Data 1–4.

## Strontium, neodymium and zinc isotopic compositions

The K-Grt cores were carefully chosen under a microscope from four garnet lherzolite samples and subsequently crushed in an agate mortar to ensure representative sampling. Due to the low Sr (<5 ppm) and Nd (<1 ppm) concentrations (Supplementary Data 3) of these decomposed garnet samples, 300 mg powder was prepared for each sample to conduct whole-rock Sr and Nd isotopic analyses. The Sr and Nd isotopic ratios were measured using a Triton thermal ionization mass spectrometer at GPMR. To facilitate comparison and interpretation, the Sr and Nd isotopic ratios were normalized to $^{86}Sr/^{88}Sr = 0.1194$ and $^{146}Nd/^{144}Nd = 0.7219$, respectively. During the analysis, the NBS987 standard yielded an average $^{87}Sr/^{86}Sr$ value of $0.710249 \pm 10$ (2σ), and the BCR-2 standard gave an average $^{143}Nd/^{144}Nd$ value of $0.512630 \pm 2$ (2σ). Details of Sr and Nd isotopic analytical procedures can be found in Li et al.[52] and Wei et al.[53]. Zn isotopic analyses were performed on a Nu Plasma 1700 MC-ICP-MS instrument, at GPMR, China University of Geosciences (Wuhan), China. Due to the small sample size, Zn isotopic analysis for sample SD53 was not performed. The data are reported in δ-notation in per mil relative to the JMC 3-0749 L: $\delta^{66}Zn$ or $\delta^{68}Zn = ((^{66}Zn/^{64}Zn)_{sample}/(^{66}Zn/^{64}Zn)_{JMC3-0749L} - 1) \times 1000$. The external reproducibility for $\delta^{66}Zn$ measurement is better than ±0.05‰ (2σ) based on long-term analyses of the international basalt standard BHVO-2 (0.28 ± 0.07‰) and standard solution NIST 682 (−2.45 ± 0.05‰). The hole-rock Sr, Nd and Zn isotopic data are presented in Supplementary Data 5.

## Transmission electron microscope (TEM) analyses

The internal structure of the olivine inclusions ($Ol_3$) was studied using an FEI Talos F200S transmission electron microscope at the Guangzhou Institute of Geochemistry, Chinese Academy of Sciences. A combination of various techniques, including scanning transmission electron microscopy (STEM), high-resolution transmission electron microscopy (HRTEM), selected area electron diffraction (SAED), and energy-dispersive X-ray spectroscopy (EDS), was used for comprehensive morphological, structural, and compositional investigations. Major elements of the $Ol_3$, along with its elliptic and octahedral inclusions, were analyzed via STEM-EDS and are presented in Supplementary Data 6. Additionally, to investigate the valence and potential coordination of Fe in the samples, spatially resolved electron energy-loss spectroscopy (EELS) was utilized at high energy resolution in the STEM mode (STEM-EELS). The EELS analyses were performed using a Gatan 1077 EELS spectrometer, with a pixel step of 2 nm for acquiring spectroscopic images. All data were acquired in a dual EELS mode with zero-peak locking. Subsequent to data acquisition, all EELS data processing tasks, including background subduction, signal integration, data fitting, and mapping, were conducted using the Gatan Microscope Suite (GMS) software (version 3.50). The assessment of $Fe^{3+}$ content in the elliptic inclusion and olivine ($Ol_3$) was conducted using a modified integral intensity ratio of the Fe $L_2$ and $L_3$ white lines with two 2.1-eV-wide integration windows ranging from 708.85 to 710.95 eV and

from 719.65 to 721.75 eV, respectively (Fig. 3d), with absolute errors estimated at approximately ±0.04 for $Fe^{3+}/\Sigma Fe$ ($\Sigma Fe = Fe^{3+} + Fe^{2+}$) ratios. For a comprehensive understanding of the $Fe^{3+}/\Sigma Fe$ calculation procedure in EELS analysis, as well as analytical precision and accuracy, readers are referred to Xian et al.[9].

## Data availability

All data generated or analyzed in this study are provided in the Supplementary Data. All data generated in this study have been deposited in the Figshare database under accession code doi.org/10.6084/m9.figshare.29144660.

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

## Acknowledgements

This study was funded by the National Natural Science Foundation of China (NSFC 42225202), the National Key Research and Development Project of China (2023YFF0804100), NSFC 41603026, and the MOST Special Fund from the State Key Laboratory of Geological Processes and Mineral Resources, China University of Geosciences (GKZ22Y607). We thank G.-L. Pan, Q. Yang, T. Chen, and J.Y. Zheng for their help in sample collection and preparation. We thank X.D. Deng, Q. Xiong, and Z.-C. Wang for helpful discussions.

## Author contributions

F.P., X.W., J.Z. conceived the project. F.P. conducted the fieldwork. F.P., X.W., J.Z., and Y.Y. interpreted the data, prepared figures, and wrote the main manuscript text. C.W., Y.Z., L.Z., C.J., X.H., H.Z. and H.H. interpreted the data and improved the manuscript. All authors contributed intellectually to the paper.

## Competing interests

The authors declare no competing interests.
