## [Transparent Peer Review file · Nature Communications]

Iron disproportionation in peridotite fragments from the mantle transition zone

Corresponding Author: Professor Junfeng Zhang

Version 0:

Reviewer comments:

Reviewer #1

(Remarks to the Author)

Recommendation: Major Revision

This manuscript presents a study on peridotite xenoliths from Cenozoic basalts in eastern China, proposing evidence for iron disproportionation in the mantle transition zone (MTZ). The authors link Na-rich majoritic garnets, Fe-Ni inclusions, and the stagnant Pacific slab to redox processes in the deep mantle. While the topic is novel and potentially impactful, the manuscript falls short of the evidentiary rigor and comprehensive analysis expected for Nature Communications.

My primary concern lies in the robustness of the central claim that the peridotite xenoliths originate from the MTZ and provide direct evidence of iron disproportionation. This claim requires significant reinforcement due to the following points:

- (1) The overwhelming majority of peridotite xenoliths worldwide are derived from the lithospheric mantle or the lithosphere-asthenosphere boundary, not the MTZ. The manuscript does not convincingly address why the studied xenoliths would be an exception, nor does it provide unequivocal evidence supporting their MTZ origin.
- (2) Prior research on peridotite xenoliths from Cenozoic basalts in eastern China predominantly attributes their origin to fragments of the subcontinental lithospheric mantle. This view contradicts the MTZ origin proposed here.
- (3) If these xenoliths originated in the MTZ, the host basalt must source from greater or equivalent depths to have capacity to entrain material from the MTZ. However, eastern China basalts are widely recognized as products of asthenospheric mantle melting at depths shallower than 300 km. This is further supported by the authors' earlier work (Fig 12 of Pan et al., 2021), which concluded that mantle melting responsible for the host basalts of the xenoliths occurred within the asthenosphere rather than the MTZ.

Specific Points to Address

- (1) Host Basalt Provenance: Basalts derived from shallower depths (<300 km) are unlikely to transport xenoliths from the MTZ. The depth of the host basalt's origin should be well re-constrained to ensure compatibility with the capture and transport of MTZ fragments.
- (2) Melting in the MTZ: The manuscript posits that melting occurs within the MTZ but does not address the lack of conventional melting triggers such as decompression, heating, or water addition in this environment. It should also clarify how MTZ-derived melts might migrate to shallower depths while entraining xenoliths.
- (3) MTZ Origin of Peridotite Xenoliths: The reliance on Na-rich majoritic garnet compositions and equilibrium pressure calculations is insufficient to conclusively establish an MTZ origin. The evidence for retrograded wadsleyite/ringwoodite derived from olivine inclusions is unconvincing. Additional evidence is required. I recommend that (1) conduct detailed comparisons with garnets from the lithospheric and asthenospheric mantle to definitively rule out shallower origins; (2) present additional structural and spectroscopic evidence to differentiate retrograded high-pressure phases from decompression-induced features.

Additional Concerns

The isotopic similarity of garnets to carbonated MORBs is interesting but insufficient to confirm slab-derived carbonate contributions. Sr-Nd isotopes alone lack the sensitivity required for such interpretations. The association of garnet isotopes with the stagnant Pacific slab remains speculative without broader comparative data. Incorporate other geochemical proxies to corroborate the proposed linkage to subducted slab components is necessary.

The manuscript does not address potential interactions between the peridotite xenoliths and the host basalt, which could significantly alter the mineralogical and geochemical signatures.

Perhaps high-pressure signatures alone do not unequivocally confirm MTZ origin. Alternative explanations, such as localized subduction-related anomalies, should be considered.

Certain technical terms, such as "kelyphitized garnet," require clearer definitions to ensure accessibility to a broader scientific audience.

Reviewer #2

(Remarks to the Author)

The manuscript „Iron disproportionation in peridotite fragments from the mantle transition zone” by Pan et al. is concerned with the discovery of (Fe,Ni) metal in olivine and kelyphitized majoritic garnet from xenoliths brought to the surface by Cenozoic basalts from eastern China. The story is that these rock fragments represent former basaltic rocks from the ocean floor that were subducted to transition zone depth (down to ~550 km depth) and later entrained in mantle upwellings bringing them back to the surface. The metal inclusions may then serve as (first) natural evidence for reducing conditions in the transition zone predicted by high P-T experiments.

The study is based on two interpretations, firstly that the garnets represent earlier oceanic crust and secondly that the garnets were subducted very deeply into the transition zone. Both interpretations should, in my opinion, be better substantiated (major revisions) before the manuscript can be published (see comments below). Nevertheless, the overall work is of very good quality, the samples are spectacular and the story is timely and certainly of interest for a broad scientific audience.

(1) Pressure derived from Garnet compositions. I agree that the high Na contents in garnet are most likely caused by high P conditions. However, I don't understand why the authors use Bobrov et al. to derive the pressure of garnet formation/equilibration (Fig. 2a). Bobrov describes his approach such that it "may be the basis for a majorite geobarometer", meaning that it is not calibrated as a geobarometer and the relationship I shown in Fig. 2a neither is. Problematic is, that in Bobrov's chemical system Na and Si contents of the resulting majorites are strongly correlated and both Na and Si contents depend on pressure. This is not the case in the present study as Na contents vary significantly at constant Si contents implying that the mechanism of Na incorporation in majoritic garnet is likely different and the P estimate thus in error. I'd recommend to use the Beyer et al. (2017) geobarometer (also in Fig. 2) to derive pressures as it is actually calibrated for Na-rich compositions and involves Na and Si.

Importantly, even if you use the Beyer et al. calibration (perhaps giving similar pressures), it remains puzzling and requires further explanation that Na and Si concentrations of the garnets are obviously decoupled in the samples presented here. To my knowledge this was never observed before. The data of Bobrov, Gasparik, Kiseeva and Beyer clearly show, that Na and Si must be correlated in Na majorites as Na is incorporated likely as Na₂M(Si₂)Si₃O₁₂ sodium-majorite (Na-Maj) component via the substitution (Mg²⁺ + Al³⁺ = Na⁺ + Si⁴⁺).

(2) Origin of the garnets. The authors claim that the garnets stem from subducted oceanic crust and belong thus to the eclogitic suite. However, the xenoliths in which they are found are lherzolitic/peridotitic. Does this imply that the garnets are eclogitic xenocrysts within the peridotitic xenoliths? How is it possible to disintegrate the eclogite and incorporate only garnet into the peridotite? Please elaborate a bit on the process you envisage for this uncommon finding.

Not sure I understand the Sr isotope argument: The garnets show high ⁸⁷Sr/⁸⁶Sr at constant ¹⁴³Nd/¹⁴⁴Nd. This is commonly attributed to seawater alteration (⁸⁷Sr/⁸⁶Sr = 0.709) of the subducted MORB which was transformed to eclogite during subduction. Is that what is meant with "carbonated MORB" in Fig. 2b? Why should it be carbonate instead of fluid and why is this signature preserved in the garnet despite dehydration/decarbonation during subduction?

Line 32: maybe delete vigorous

Line 49: maybe give a real number of samples?

Line 51: "... of the continent..."

Line 50-55: unclear, please consider rephrasing (rocks migrate oceanward?)

Line 66: What is the composition of the Fe,Ni alloy? It would be very interesting to know this composition to constrain the fO₂ further and to correlate the alloy composition with the model of Frost and McCammon 2008 (Ni-rich close to the Ni-precipitation curve and becoming progressively more Fe rich with increasing pressure). The alloys were measured by EELS but not by TEM-EDX or EPMA?

Line 79: What is meant by Na-rich matrix and why wasn't it characterized further using e.g., TEM? It seems to be a substantial part of the former garnet core so any breakdown reaction should involve this Na-rich matrix, I guess.

Line 83: lithophile elements? Please explain a bit more detailed what Ext. Data Fig 4 tells about the garnets. 4b reflects the partitioning behavior for REE but what is the implication of Fig 4a is and why is it indicative for "...minimal exchange with external fluids/melts..." (line 84-85).

Line 86: This is important so I'd suggest to present this in greater detail and also cite some relevant literature here, for example the work of Obata et al. regarding the textures.

Line 113: please put references here.

Line 117: synthetic eclogite?

Lines 185-187: Any reference for the method? Van Aken 1998 I guess or is this implemented in the Gatan software?

Lines 192: maybe add Schmidt et al. Progress in Earth and Planetary Science 2014, 1:27 as reference for SiC originating from reduced fluids

Line 195: I'd suggest to consider only Fe²⁺ for calculating the #Mg.

Lines 203ff: I do not really understand the arguments here. The olivine structure has no position for Fe³⁺, that's why Fe³⁺ is so low in olivine in all geologic settings. Even if high-P wadsleyite contained Fe³⁺, the retrograde back transformation to olivine should exclude all Fe³⁺ from this olivine. There seems no way that olivine hosts 40% (3-4 wt.%) structural Fe₂O₃. Please elaborate how this should be possible. Further, I don't understand why hydrous experimental wadsleyite is considered for comparison as interaction with fluids is excluded as likely scenario.

Line 257: what is meant by RHP? Was the Ni-content measured in the Fe,Ni phases?

Line 405: kV

Line 463: chosen

Line 494: Σ

Overall very nice work! With best wishes,

Version 1:

Reviewer comments:

Reviewer #1

(Remarks to the Author)

The authors have addressed all my concerns and provided comprehensive responses to my comments. The manuscript has improved significantly in terms of clarity and quality. However, I still believe that the first issue I raised (MTZ origin for the xenoliths) is worth further investigation. Providing additional evidence, such as other compositional information from MTZ sources, would strengthen the argument and enhance the manuscript's persuasiveness.

That said, the paper is internally consistent and well-structured. I recommend publication either in its current form or after minor revisions to further refine the discussion.

Reviewer #2

(Remarks to the Author)

The revised version of "Iron disproportionation in peridotite fragments from the mantle transition zone" is more to the point and addresses the questions/aspects I raised in the review properly, either in the main text or in the rebuttal. Specifically, the authors use now the Beyer geobarometer to calculate pressures, provide an explanation for the decoupling of Na and Si in the garnet, clarify how eclogitic garnets are incorporated in lherzolitic xenoliths provide an explanation how Fe³⁺ is incorporated in olivine. Also, in my opinion, the concerns of reviewer 1 are adequately addressed but I'd leave that decision to the other reviewer or the editors, of course. Overall, I'd be happy to see this manuscript being published in Nature Communications, to me it is high quality work on an exciting topic. I have some additional minor comments the authors might adopt.

1) Maybe elaborate a bit on the uncertainties of the pressure calculation. Given the rather large variation for the Na measurements in garnet, the reader might be interested how this translates into a 2 sigma uncertainty of P.

2) Looking again at the garnet compositions and on the criteria to distinguish eclogitic from peridotitic garnet (Fig. 7 of Beyer and Frost), it seems that garnets M1-M7 are rather (besides the Na content) transitional in Cr content and Ca# and belong clearly to the peridotitic suite in terms of Mg# and Ti content. Maybe it would be interesting for the reader to address this apparent contradiction.

3) Fig. 2 in the rebuttal letter is very nice, maybe consider to put it into the Supplement.

4) Line 103 (file with track changes) lithophile

5) Line 135 (file with track changes) I'd rather say "calculated" and give uncertainties

Responses to the comments from the reviewers (in blue color)

Reviewer #1 (Remarks to the Author):

Recommendation: Major Revision

This manuscript presents a study on peridotite xenoliths from Cenozoic basalts in eastern China, proposing evidence for iron disproportionation in the mantle transition zone (MTZ). The authors link Na-rich majoritic garnets, Fe-Ni inclusions, and the stagnant Pacific slab to redox processes in the deep mantle. While the topic is novel and potentially impactful, the manuscript falls short of the evidentiary rigor and comprehensive analysis expected for Nature Communications.

My primary concern lies in the robustness of the central claim that the peridotite xenoliths originate from the MTZ and provide direct evidence of iron disproportionation. This claim requires significant reinforcement due to the following points:

- (1) The overwhelming majority of peridotite xenoliths worldwide are derived from the lithospheric mantle or the lithosphere-asthenosphere boundary, not the MTZ. The manuscript does not convincingly address why the studied xenoliths would be an exception, nor does it provide unequivocal evidence supporting their MTZ origin.
- (2) Prior research on peridotite xenoliths from Cenozoic basalts in eastern China predominantly attributes their origin to fragments of the subcontinental lithospheric mantle. This view contradicts the MTZ origin proposed here.
- (3) If these xenoliths originated in the MTZ, the host basalt must source from greater or equivalent depths to have capacity to entrain material from the MTZ. However, eastern China basalts are widely recognized as products of asthenospheric mantle melting at depths shallower than 300 km. This is further supported by the authors' earlier work (Fig 12 of Pan et al., 2021), which concluded that mantle melting responsible for the host basalts of the xenoliths occurred within the asthenosphere rather than the MTZ.

Response: Thank you for your thoughtful comments. Tracing the depth origin of mantle xenoliths is indeed a challenging task due to the decomposition of high-pressure deep mantle minerals during their ascent to the surface. Most mineralogical and geochemical evidence that could help trace their origin tends to be either erased or significantly altered during upwelling. While it is true that the majority of peridotite xenoliths worldwide are thought to derive from the lithospheric mantle or the lithosphere-asthenosphere boundary, our xenoliths are unique in several ways that support the MTZ origin.

1. **Garnet vs. Spinel Peridotites:** The majority of previous studies on peridotite xenoliths from eastern China have reported spinel peridotites. In contrast, our study reports garnet peridotites, which exhibit distinctly different mineral associations and geochemical compositions. This is a clear indication that these xenoliths are not typical lithospheric

peridotites.

2. **Geochemical Evidence of the host rock.** The mantle source of the Cenozoic intraplate OIB-like basalts in eastern China has been shown to involve recycled carbonated slab components from the MTZ. This interpretation is supported by accumulating evidence from major- and trace-elements geochemistry, Sr-Nd-Pb-Zn-Mg-Ca-O isotopic compositions (e.g., Dong et al., 2024; Li et al., 2017; Pan et al., 2021; Wang et al., 2015; Zeng et al., 2021). Moreover, recent studies have suggested that the contemporaneous nephelinites in eastern China, which resemble the host rock of our studied garnet peridotites, also originate from the MTZ (Zeng et al., 2021).
3. **Direct Evidence for MTZ Origin.** We provided three key pieces of evidence supporting the MTZ origin of these garnet peridotite xenoliths. 1) Pressure-induced $\text{Fe}^{2+} \rightarrow \text{Fe}^0 + \text{Fe}^{3+}$ disproportionation in silicates. This phenomenon, which is typical of deep mantle below 250 km (Forst et al., 2004; Rohrbach et al., 2007), was observed in kelyphitized Na-rich majoritic garnet and olivine. 2) Na-rich majoritic garnet. Known to be a reliable pressure indicator for deep mantle materials at depths of 300-660 km (Beyer and Frost, 2017; Bobrov et al., 2008), we determined formation pressures (18.9–13.3 GPa) based on the sodium and potassium contents in the decomposed garnet. 3) Mioré Patterns of spinel + olivine + Fe nano-inclusions. These patterns in olivine point to a high-pressure origin, consistent with minerals found in the MTZ, such as wadsleyite/ringwoodite.

These three lines of evidence are self-consistent and provide robust support for the MTZ origin of the xenoliths, a finding that has not been reported previously in mantle xenoliths from eastern China.

1. **Host Basalt Provenance:** Basalts derived from shallower depths (<300 km) are unlikely to transport xenoliths from the MTZ. The depth of the host basalt's origin should be well re-constrained to ensure compatibility with the capture and transport of MTZ fragments.

Response: The host rock for the studied xenoliths is a nephelinite, whose major and trace elements are provided in Supplementary Table 1. The contemporaneous nephelinites in eastern China, which closely resemble the host rock of our garnet peridotite samples, have also been suggested to originate from the MTZ (Zeng et al., 2021). To address your concern, we have rephrased the text to include information on the composition of the host rock (lines 52-58).

Additionally, considerable portions of the mantle below 250 km are saturated with an iron–nickel rich metal phase formed by the ferrous disproportionation of iron-bearing silicates, with the redox state controlled by the iron–wüstite (IW) buffer (Forst et al., 2004; Rohrbach et al., 2007; Kuwahara et al., 2023). As oxygen fugacity in Earth's interior decreases with increasing depth, the ferrous disproportionation found in this study provides direct mineralogical evidence of a deep origin (more than 250 km depth at least) of the host nephelinites.

2. **Melting in the MTZ:** The manuscript posits that melting occurs within the MTZ but does not

address the lack of conventional melting triggers such as decompression, heating, or water addition in this environment. It should also clarify how MTZ-derived melts might migrate to shallower depths while entraining xenoliths.

Response: An intermittent layer composed of low seismic wave velocity zones (LVZs) has been identified globally just above the top of the mantle transition zone (e.g., Revenaugh et al., 1994; Tauzin and Debayle, 2010). These LVZs are commonly attributed to a small amount of silicate-rich melts in the mantle (Revenaugh et al., 1994; Tauzin and Debayle, 2010) or grain-size sensitive anelastic relaxation near the mantle solidus (Karato, 2012; Takei, 2017). High-temperature and high-pressure partial melting experiments have found that carbon redox melting plays an important role in deep mantle melting (Dasgupta et al., 2013; Rohrbach and Schmidt, 2011), which is regulated by the mantle oxygen fugacity depended on the ferric iron content relative to total iron ($\text{Fe}^{3+}/\Sigma\text{Fe}$) (Moussallam et al., 2019; Stagno et al., 2013; Rohrbach and Schmidt, 2011). For a typical asthenosphere source, a $\text{Fe}^{3+}/\Sigma\text{Fe}$ ratio of nearly 3-4% leads to carbonated silicate melt generation at 150-250 km depth (Dasgupta et al., 2013; Stagno et al., 2013). However, when the whole-rock ferric iron content exceeds 12% of total Fe, carbonated melting would commence at depths >300 km (Moussallam et al., 2019; Stagno et al., 2013).

LVZs are widely found atop the mantle transition zone (300-400 km) in eastern China, with a thickness ranging from 50-80 km in North China to 20-60 km in South China (Tauzin et al., 2010; Ma et al., 2020; Han et al., 2021). The widespread Cenozoic intraplate volcanisms in eastern China are spatially distributed above the LVZs, indicating a potential petrogenesis contribution from the MTZ, as suggested by geophysical modeling results (Yang and Faccenda, 2020). Geochemical data on these intraplate basalts also speculated that some of them may derive from the MTZ (Wang et al., 2015; Zeng et al., 2021). The metal inclusions found in this study provided first direct natural evidence for reducing conditions in the transition zone.

The ferrous disproportionation observed could also alter the relative concentrations of Fe^{3+} , Fe^{2+} , and Fe^0 , potentially regulating oxygen fugacity and inducing partial melting in the deep mantle. We have identified a garnet pyroxenite (Figure 1) that may serve as a highly oxidized magma source for deep (> 300 km) upper mantle melting. This pyroxenite xenolith contains a significant proportion (~15%) of Na-rich majoritic garnet. Olivine clinopyroxenite (Region A) and garnet olivine websterite (Region C) show an interbedded structure (Figure 1a). Fe-Ni alloys are present in some kelyphitized Na-rich majoritic garnets, and Fe-Cu-Ni-rich sulfides are present in the nepheline veins of Region C (Figures 1f and 1g), indicating a highly reduced condition. Sulfate veins, sulfide grains and graphite (Figures 1d and 1e) are found at the contact interface between olivine clinopyroxenite and garnet olivine websterite (Region B). The olivine clinopyroxenite part was sampled for whole-rock major element analysis, revealing the following composition: SiO_2 of 50.53 wt.%, Al_2O_3 of 7.43 wt.%, CaO of 13.70 wt.%, MgO of 19.26 wt.%, and TFe_2O_3 of 4.65 wt.% with Mg# value of 89. This olivine clinopyroxenite is also characterized by a high ferric iron content ($\text{Fe}^{3+}/\Sigma\text{Fe} = 0.12$). The distinct oxygen fugacity signatures preserved in the garnet pyroxenite, and their potential effects on partial melting of the upwelling asthenosphere, will be discussed in our subsequent papers. In this study, we aim to

provide mineralogical evidence (such as ferrous disproportionation in olivine or its high pressure polymorphs) for the deep origin of these unique mantle xenoliths.

Figure 1. Inverse oxygen fugacity recorded in a pyroxenite xenolith. (a) The pyroxenite consists of three distinct regions: garnet-free pyroxenite (Region A), garnet-rich pyroxenite (Region C), and the interaction zone between them (Region B). (b) and (c) Region A is composed of coarse-grained anhedral orthopyroxene (Opx) and clinopyroxene (Cpx I), as well as fine-grained clinopyroxene (Cpx II), olivine (Ol), and nepheline (Nph). The fine-grained Ol and Cpx II are interpreted to have formed through reactions between Opx and carbonatitic melts. (d) and (e) Region B shows the coexistence of oxidizing sulfate (Sft) and reducing sulfide and graphite (Gp), indicative of different oxygen fugacity conditions. (f) and (g) Region C is characterized by coarse-grained garnet (Grt), Opx, and Cpx I. Reducing sulfides and nepheline are common in the mineral assemblage, and Fe-Ni alloys are observed in the garnet cores.

3. MTZ Origin of Peridotite Xenoliths: The reliance on Na-rich majoritic garnet compositions and equilibrium pressure calculations is insufficient to conclusively establish an MTZ origin. The evidence for retrograded wadsleyite/ringwoodite derived from olivine inclusions is unconvincing. Additional evidence is required. I recommend that (1) conduct detailed comparisons with garnets from the lithospheric and asthenospheric mantle to definitively rule out shallower origins; (2) present additional structural and spectroscopic evidence to differentiate retrograded high-pressure phases from decompression-induced features.

Response: The reviewer didn't provide specific reasons for dismissing the evidence we presented. The majorite component can only exist in garnets that equilibrated at depth > 150 km, as a result of the dissolution of coexisting orthopyroxene and clinopyroxene into garnet (Ringwood and Major, 1971; Beyer et al., 2017). The Na-rich majorite component is in equilibrium at even higher pressures, above 7 GPa (Bobrov et al., 2008a, b; Dymshits et al., 2013). Thus, the high-pressure crystal structures of most majoritic garnets are typically preserved as inclusions in diamonds (Kiseeva et al., 2013 and references therein).

Wadsleyite and ringwoodite are dominant minerals in the mantle transition zone (MTZ). These high-pressure polymorphs of olivine are rarely preserved in deep mantle diamonds (Pearson et al., 2014; Gu et al., 2022). Our peridotite xenolith samples were captured in an open igneous system, which makes it extremely difficult to preserve the original phases of these ultra-high-pressure minerals. The Na-rich matrix in the decomposed garnet is dominantly composed of plagioclase crystallites with varying Na₂O, K₂O, and CaO contents (Supplementary Figs. 3 and 4). This submicron structure precludes an exotic origin for the alkali contents. Therefore, the high Na₂O (0.75–2.41 wt.%) and K₂O (0.11–0.73 wt.%) signatures observed in the studied Na-rich majoritic garnets strongly support a retrograded high-pressure phase origin.

The presence of Fe³⁺-rich ($Fe^{3+}/\Sigma Fe = 0.35-0.40$) olivine and widespread spherical Fe-Ni alloys further confirms pressure-driven disproportionation of iron in silicates in the deep mantle. Iron-saturation due to pressure-driven ferrous disproportionation is widely accepted to occur only at depths > 250 km (Rohrbach et al., 2007; Frost et al., 2008; Beyer et al., 2021; Armstrong et al., 2019; Kuwahara et al., 2023), which rules out a shallow origin for the Na-rich kelyphitized garnets and the olivine inclusions we studied.

High-resolution transmission electron microscope (HRTEM) images and corresponding fast Fourier transform (FFT)\nano-beam electron diffraction (NBD) reveal comparable Mioré Patterns between the octahedral inclusion and elliptic inclusions (Extended Data Fig. 7m-o). We also determined their compositions using STEM-EDS. Based on their structural characteristics and compositions, we conclude that these inclusions are a mixture of olivine (Ol₄) and spinel. The residual grains are smaller than 100 nm, which is too small for spectroscopic analysis.

Supplementary Figure 4. Transmission electron microscopy analyses of the kelyphitized garnet core. Panel (a) is a HAADF-STEM micrograph, while panels (b-k) are EDS elemental maps.

4. The isotopic similarity of garnets to carbonated MORBs is interesting but insufficient to confirm slab-derived carbonate contributions. Sr-Nd isotopes alone lack the sensitivity required for such interpretations. The association of garnet isotopes with the stagnant Pacific slab remains speculative without broader comparative data. Incorporate other geochemical proxies to corroborate the proposed linkage to subducted slab components is necessary.

Response: Carbonated signatures from recycled subducted slab components have been consistently proposed as contributing to the mantle source of the Cenozoic intraplate OIB-like basalts in eastern China, based on major- and trace-elements geochemistry, as well as Sr-Nd-Pb-Zn-Mg-Ca-O isotopic compositions (i.e. Dong et al., 2024; Li et al., 2017; Pan et al., 2021; Wang et al., 2015; Zeng et al., 2021). The Sr-Nd isotopic compositions of the Na-rich majoritic garnet are compared with those of asthenosphere-derived magmas, such as MORBs,

HIUM, EM-1, EM-2 (Fig. 2b).

Following the reviewer's suggestion, we have also measured Zn isotopes in the Na-rich majoritic garnets and compared them with those of the Cenozoic intraplate basalts in eastern China (supplementary Fig. 6c). The isotopic similarity between the garnets and the basalts further supports the hypothesis of an association with the stagnant Pacific slab in the MTZ. These new data have been included in the revised text (lines 155-160).

Supplementary Figure 6. Characteristic compositions and the Zn-Sr isotopes of the decomposed Na-rich majoritic garnets. (a) Plot of the sum of ($Mg^{2+} + Ca^{2+} + Fe^{2+} + Mn^{2+}$) p.f.u. versus $Si + Ti$ (p.f.u.). (b) Plot of the sum of ($Al^{3+} + Cr^{3+}$) p.f.u. versus $Si + Ti$ (p.f.u.). Majoritic garnet inclusions from natural diamonds are sourced from Kiseeva et al. (2013). The experimental data (Jd) are from Bobrov et al. (2008). All Fe in majoritic garnet is treated as Fe^{2+} for comparison. (c) Zn and Sr isotopic compositions of the decomposed Na-rich majoritic garnets and Cenozoic intraplate basalts from our study area (grey rhombus). The Zn-Sr isotopic compositions of the decomposed garnets are consistent with those of carbonated eclogite. Both the Zn-Sr isotopes of the Cenozoic intraplate basalts and the mixing model are according to Xu et al. (2022).

5. The manuscript does not address potential interactions between the peridotite xenoliths and the host basalt, which could significantly alter the mineralogical and geochemical signatures.

Response: Thank you for this excellent point. The host rock for the studied xenoliths is nephelinite, and the major and trace elements of this rock have been provided in Supplementary Table 1. The ferric iron contents ($Fe^{3+}/\Sigma Fe$) of the nephelinite, analyzed by wet chemistry measurements, range from 0.25 to 0.31, which is consistent with the highly oxidized Cenozoic intraplate basalts in eastern China (Dong et al., 2024). Additionally, Fe-Ni alloys are widespread in both the garnet core and olivine (Ol_3), which precludes significant interactions between the peridotite xenoliths and the highly oxidized host nephelinite. Furthermore, the decomposed Na-rich majoritic garnet shows typical garnet geochemical signatures that clearly differ from those of the host nephelinite (Supplementary Figure 5 in the Extended Data). These observations strongly suggest that interactions between the xenoliths and the host nephelinite were negligible.

We have rephrased the text for clarity (lines 82-84 and 94-99).

6. Perhaps high-pressure signatures alone do not unequivocally confirm MTZ origin. Alternative explanations, such as localized subduction-related anomalies, should be considered.

Response: This is precisely the point we are making in the manuscript. As shown by the geophysical observations (Fig. 4 & Wei et al., 2012), the most significant subduction-related anomalies in the mantle region are the stagnant slab in the MTZ. To our best knowledge, there are currently no reports of other subduction-related anomalies in the upper mantle beneath eastern China.

7. Certain technical terms, such as "kelyphitized garnet," require clearer definitions to ensure accessibility to a broader scientific audience.

Response: Thank you for the suggestion. We have clarified the two kelyphitization mechanisms in the revised manuscript (lines 72-102). The kelyphitized garnet rim forms due to rapid exhumation via the reaction between garnet and olivine ($\text{Grt} + \text{Ol}_1 = \text{Opx} + \text{Cpx} + \text{Sp/Pl}$), with plagioclase representing the terminal phase of this retrograde reaction. Meanwhile, the K-Grt core is likely the isostoichiometric breakdown product of the parental garnet itself ($\text{Grt} = \text{Opx} + \text{Sp} + \text{An} \pm \text{Cpx}$).

Reviewer #2 Evaluations:

The manuscript "Iron disproportionation in peridotite fragments from the mantle transition zone" by Pan et al. is concerned with the discovery of (Fe,Ni) metal in olivine and kelyphitized majoritic garnet from xenoliths brought to the surface by Cenozoic basalts from eastern China. The story is that these rock fragments represent former basaltic rocks from the ocean floor that were subducted to transition zone depth (down to ~550 km depth) and later entrained in mantle upwellings bringing them back to the surface. The metal inclusions may then serve as (first) natural evidence for reducing conditions in the transition zone predicted by high P-T experiments.

The study is based on two interpretations, firstly that the garnets represent earlier oceanic crust and secondly that the garnets were subducted very deeply into the transition zone. Both interpretations should, in my opinion, be better substantiated (major revisions) before the manuscript can be published (see comments below). Nevertheless, the overall work is of very good quality, the samples are spectacular and the story is timely and certainly of interest for a broad scientific audience.

1. Pressure derived from Garnet compositions. I agree that the high Na contents in garnet are most likely caused by high P conditions. However, I don't understand why the authors use Bobrov et al. to derive the pressure of garnet formation/equilibration (Fig. 2a). Bobrov describes his approach such that it "may be the basis for a majorite geobarometer", meaning

that it is not calibrated as a geobarometer and the relationship I shown in Fig. 2a neither is. Problematic is, that in Bobrov's chemical system Na and Si contents of the resulting majorites are strongly correlated and both Na and Si contents depend on pressure. This is not the case in the present study as Na contents vary significantly at constant Si contents implying that the mechanism of Na incorporation in majoritic garnet is likely different and the P estimate thus in error. I'd recommend to use the Beyer et al. (2017) geobarometer (also in Fig. 2) to derive pressures as it is actually calibrated for Na-rich compositions and involves Na and Si.

Importantly, even if you use the Beyer et al. calibration (perhaps giving similar pressures), it remains puzzling and requires further explanation that Na and Si concentrations of the garnets are obviously decoupled in the samples presented here. To my knowledge this was never observed before. The data of Bobrov, Gasparik, Kiseeva and Beyer clearly show, that Na and Si must be correlated in Na majorites as Na is incorporated likely as $\text{Na}_2\text{M}(\text{Si}_2)\text{Si}_3\text{O}_{12}$ sodium-majorite (Na-Maj) component via the substitution ($\text{Mg}^{2+} + \text{Al}^{3+} = \text{Na}^{+} + \text{Si}^{4+}$).

Response: Thank you for the constructive feedback. We agree with your point regarding the use of the Bobrov et al. (2008) approach to derive the pressure of garnet formation/equilibration. We have taken your suggestion and revised our manuscript to use the Beyer et al. (2017) geobarometer to estimate the pressures, as it is specifically calibrated for Na-rich compositions and accounts for both Na and Si contents. We have updated Figure 2 accordingly and discussed the application of the Beyer et al. geobarometer in the revised text.

Regarding the decoupling of Na and Si concentrations in our garnets, we hypothesize that this could be related to the kelyphitization process, which occurred in a relatively closed isostoichiometric system with minimal exchange with external fluids/melts. However, the presence of plagioclase in the garnet rim suggests that sodium and potassium were lost from the garnet core (K-Grt) to the rim, leading to the observed decoupling. We originally discussed this potential influence in the "Analytical Methods" section, but we have now moved this discussion to the main text (lines 114-119) for clarity.

We appreciate your valuable input and hope these revisions address your concerns effectively.

2. Origin of the garnets. The authors claim that the garnets stem from subducted oceanic crust and belong thus to the eclogitic suite. However, the xenoliths in which they are found are lherzolitic/peridotitic. Does this imply that the garnets are eclogitic xenocrysts within the peridotitic xenoliths? How is it possible to disintegrate the eclogite and incorporate only garnet into the peridotite? Please elaborate a bit on the process you envisage for this uncommon finding.

Not sure I understand the Sr isotope argument: The garnets show high $^{87}\text{Sr}/^{86}\text{Sr}$ at constant $^{143}\text{Nd}/^{144}\text{Nd}$. This is commonly attributed to seawater alteration ($^{87}\text{Sr}/^{86}\text{Sr} = 0.709$) of the subducted MORB which was transformed to eclogite during subduction. Is that what is

meant with “carbonated MORB” in Fig. 2b? Why should it be carbonate instead of fluid and why is this signature preserved in the garnet despite dehydration/decarbonation during subduction?

Response: Thank you for your thoughtful comments and for raising important questions regarding the origin of the garnets. You are correct that the xenoliths in which the garnets are found are peridotitic, specifically lherzolitic in composition. The presence of eclogitic garnet in these peridotitic xenoliths, which are typically dominated by olivine and pyroxenes, indeed raises an interesting question. Our interpretation is that these garnets likely represent xenocrysts from a subducted oceanic crust that were captured by the surrounding peridotites during mantle upwelling processes (lines 276-283).

As oceanic crust subducts and descends into the mantle, the volume fraction of garnet increases with depth as clinopyroxene is dissolved into garnet, leading to the formation of monomineralic majoritic garnetite at pressures greater than 14 GPa in the mantle transition zone (MTZ). These majoritic garnetites may then become part of the xenoliths captured by upwelling peridotites in the MTZ, forming a mixture of wadsleyite/ringwoodite/olivine and garnetite. It is possible for these garnetites to remain as inclusions or xenocrysts within the surrounding peridotitic material after exhumation. We have rephrased the manuscript to clarify this interpretation.

Regarding the Sr isotopic compositions, we agree that high $^{87}\text{Sr}/^{86}\text{Sr}$ ratios at constant $^{143}\text{Nd}/^{144}\text{Nd}$ are typically attributed to seawater alteration of subducted oceanic crust. The “carbonated MORB” signature in Fig. 2b refers to the alteration of the subducted oceanic crust by carbon-rich fluids, likely originating from carbonate-rich sediments or altered oceanic crust itself. This process could explain the elevated Sr isotopic ratio, and we propose that it reflects the incorporation of carbonated material into the garnet during subduction.

You raise an excellent point about the preservation of this signature despite the dehydration and decarbonation that occurs during subduction. We suggest that the carbonation signature may have been retained in the garnet during its formation at high pressures in the MTZ, possibly due to the stability of carbonate in the deeper parts of the mantle. Additionally, we have analyzed the Zn isotopic compositions of the decomposed Na-rich majoritic garnet, and the Zn isotopic data show a similar pattern to that observed in Cenozoic intraplate basalts from eastern China, further supporting the hypothesis of carbonate alteration in the subducted oceanic crust.

We have updated the manuscript to incorporate these clarifications and additional data (Zn isotopes) to strengthen the argument for a carbonated origin of the garnet.

3. Line 32: maybe delete vigorous

Response: We have removed it as suggested.

4. Line 49: maybe give a real number of samples?

Response: We have now specified the number of samples in the revised text. We collected more than twenty “garnet” lherzolite xenoliths, but only five samples (SD07, SD50, SD51, SD52, and SD53) are presented in this study.

5. Line 51: “... of the continent...”

Response: We have corrected it.

6. Line 50-55: unclear, please consider rephrasing (rocks migrate oceanward?)

Response: This sentence has been rephrased as follows: “*The eruption ages of these rocks decrease oceanward, with an ~100-km spatial gap,*”

7. Line 66: What is the composition of the Fe,Ni alloy? It would be very interesting to know this composition to constrain the fO_2 further and to correlate the alloy composition with the model of Frost and McCammon 2008 (Ni-rich close to the Ni-precipitation curve and becoming progressively more Fe rich with increasing pressure). The alloys were measured by EELS but not by TEM-EDX or EPMA?

Response: The compositions of the Fe-Ni alloys have been measured by STEM-EDS (Supplementary Table 6), and we have provided additional information in revised text (lines 292-297).

8. Line 79: What is meant by Na-rich matrix and why wasn't it characterized further using e.g., TEM? It seems to be a substantial part of the former garnet core so any breakdown reaction should involve this Na-rich matrix, I guess.

Response: Following your suggestion, we have characterized the Na-rich matrix in more detail using TEM. The results show that the Na-rich matrix in the decomposed garnet consists primarily of plagioclase crystallites with variable Na_2O , K_2O , and CaO contents (Supplementary Fig. 4). We have included this new information in the revised text (lines 82-84) for better clarity.

9. Line 83: lithophile elements? Please explain a bit more detailed what Ext. Data Fig 4 tells about the garnets. 4b reflects the partitioning behavior for REE but what is the implication of Fig 4a is and why is it indicative for “...minimal exchange with external fluids/melts...” (line 84-85).

Response: Regarding your question on lithophile elements and the trace element data, we have clarified this point in the revised manuscript. The conclusion of minimal exchange with external fluids/melts is based on a comparison of the trace element compositions between the kelyphitized garnet core and their nephelinite host. As shown in Supplementary Fig. 5a and

Supplementary Table 3, the nephelinite host shows a significant enrichment in large-ion lithophile elements (LILE), typical of Cenozoic OIB-type intraplate basalts from eastern China. In contrast, the kelyphitized garnet cores exhibit extremely low concentrations of lithophile elements and are enriched in heavy rare earth elements (HREE), which is characteristic of lithospheric garnets. This significant difference in trace element signatures suggests that the garnets have undergone minimal exchange with external fluids or melts. We have rephrased the text to highlight this conclusion more clearly (lines 94-99).

10. Line 86: This is important so I'd suggest to present this in greater detail and also cite some relevant literature here, for example the work of Obata et al. regarding the textures.

Response: Following your suggestion, we have expanded the discussion (lines 84-102) to include more details and relevant references of Špaček et al. (2013) and Obata et al. (2013), which address textures related to high-pressure garnets and their decomposition.

11. Line 113: please put references here.

Response: We have cited Beyer & Frost (2017) here.

12. Line 117: synthetic eclogite?

Response: We have changed it.

13. Lines 185-187: Any reference for the method? Van Aken 1998 I guess or is this implemented in the Gatan software?

Response: van Aken et al. (1998) has been cited in the revised text as the reference for the method.

14. Lines 192: maybe add Schmidt et al. Progress in Earth and Planetary Science 2014, 1:27 as reference for SiC originating from reduced fluids

Response: This reference has been cited in the revised text.

15. Line 195: I'd suggest to consider only Fe²⁺ for calculating the #Mg.

Response: We have considered the lattice substitution of ferric iron and its implications for calculating the Mg#. Ferric iron is associated with crystal defects in olivine structures (see our response to Comment #16). Therefore, we have calculated the Mg# using total iron as the minimum value for olivine and its high-pressure polymorphs in the deep mantle.

16. Lines 203ff: I do not really understand the arguments here. The olivine structure has no position for Fe^{3+} , that's why Fe^{3+} is so low in olivine in all geologic settings. Even if high-P wadsleyite contained Fe^{3+} , the retrograde back transformation to olivine should exclude all Fe^{3+} from this olivine. There seems no way that olivine hosts 40% (3-4 wt.%) structural Fe_2O_3 . Please elaborate how this should be possible. Further, I don't understand why hydrous experimental wadsleyite is considered for comparison as interaction with fluids is excluded as likely scenario.

Response: We appreciate your insights. The argument for Fe^{3+} presence in olivine under deep mantle conditions is based on its disequilibrium in the crystal structure, where it accumulates primarily around Fe-Ni alloys, which contain numerous crystal defects (see Figure 2). These defects are likely the result of ferrous disproportionation. While ferric iron in olivine is unusual under low pressure conditions and tends to migrate out during the ascent, we hypothesize that Fe^{3+} is trapped around crystal defects associated with the Fe-Ni alloys.

As for the RHP-OI (retrograded high-pressure polymorphs of wadsleyite/ringwoodite), these grains were initially captured in the mantle transition zone (MTZ) and may have contained significant hydrogen concentrations (up to 1.5 wt.% H_2O , as reported by Fei et al., 2020, 2021; Pearce et al., 2014). This hydrogen could elevate the concentration of Fe^{3+} in wadsleyite/ringwoodite (McCammon et al., 2004; Muir et al., 2023). We also note that no evidence suggests interaction with external Na-rich melts or fluids, as previously explained.

Figure 2. Fe-Ni alloys and associated crystal defects in olivine (Ol_3). (a) and (b) show crystal defects surrounding the Fe-Ni alloys in the olivine.

17. Line 257: what is meant by RHP? Was the Ni-content measured in the Fe,Ni phases?

Response: Thank you for the clarification request. RHP refers to retrograded high-pressure polymorph of wadsleyite/ringwoodite. We have defined this term clearly in the revised text (lines 282).

Additionally, the Ni contents of the Fe-Ni phases were measured by STEM-EDS for both the Na-rich majoritic garnets and the RHP-Olivine (RHP-OI) phases. These measurements are presented in the revised text (lines 292-297) and in Supplementary Table 6.

18. Line 405: kV

Response: We have corrected it as suggested.

19. Line 463: chosen

Response: We have corrected it as suggested.

20. Line 494: Σ

Response: We have corrected it as suggested.

Cited references

Beyer, C. & Frost, D. J. The depth of sub-lithospheric diamond formation and the redistribution of carbon in the deep mantle. *Earth and Planetary Science Letters* **461**, 30-39, doi:<https://doi.org/10.1016/j.epsl.2016.12.017> (2017).

Bobrov, A. V., Kojitani, H., Akaogi, M. & Litvin, Y. A. Phase relations on the diopside – jadeite – hedenbergite join up to 24 GPa and stability of Na-bearing majoritic garnet. *Geochimica et Cosmochimica Acta* **72**, 2392-2408, doi:<https://doi.org/10.1016/j.gca.2008.03.003> (2008)

Dong, X.-H. *et al.* Highly oxidized intraplate basalts and deep carbon storage. *Science Advances* **10**, eadm8138 (2024).

Ezad, I. S. *et al.* Kelyphite textures experimentally reproduced through garnet breakdown in the presence of a melt phase. *Journal of Petrology* **63**, doi:10.1093/petrology/egac110 (2022).

Fei, H. & Katsura, T. High water solubility of ringwoodite at mantle transition zone temperature. *Earth and Planetary Science Letters* **531**, 115987 (2020).

Fei, H. & Katsura, T. Water solubility in Fe - bearing wadsleyite at mantle transition zone temperatures. *Geophysical Research Letters* **48**, e2021GL092836 (2021).

Frost, D. J. & McCammon, C. A. The redox state of Earth's mantle. *Annual Review of Earth and Planetary Sciences* **36**, 389-420, doi:10.1146/annurev.earth.36.031207.124322 (2008).

Han, G. *et al.* Pervasive low-velocity layer atop the 410-km discontinuity beneath the northwest Pacific subduction zone: Implications for rheology and geodynamics. *Earth and Planetary Science Letters* **554**, 116642, doi:<https://doi.org/10.1016/j.epsl.2020.116642> (2021).

Karato, S.-i. On the origin of the asthenosphere. *Earth and Planetary Science Letters* **321**,

95-103 (2012).

- Kiseeva, E. S. et al. Metapyroxenite in the mantle transition zone revealed from majorite inclusions in diamonds. *Geology* **41**, 883-886, doi:10.1130/g34311.1 (2013)
- Kiseeva, E. S. et al. Oxidized iron in garnets from the mantle transition zone. *Nature Geoscience* **11**, 144-147, doi:10.1038/s41561-017-0055-7 (2018).
- Kuwahara, H., Nakada, R., Kadoya, S., Yoshino, T. & Irifune, T. Hadean mantle oxidation inferred from melting of peridotite under lower-mantle conditions. *Nature Geoscience* **16**, 461-465, doi:10.1038/s41561-023-01169-4 (2023).
- Ma, M. et al. The melt content of the low velocity layer in the eastern South China: Implications for the subduction process of the Western Pacific plate. *Physics of the Earth and Planetary Interiors* **298**, 106321, doi:https://doi.org/10.1016/j.pepi.2019.106321 (2020).
- McCammon, C. et al. Oxidation state of iron in hydrous mantle phases: implications for subduction and mantle oxygen fugacity. *Physics of the Earth and Planetary interiors* **143**, 157-169 (2004).
- Muir, J. M. R., Jollands, M. & Zhang, F. The oxidation states of Iron in dry and wet olivine: a thermodynamic model. *Journal of Geophysical Research: Solid Earth* **128**, e2023JB026840, doi:https://doi.org/10.1029/2023JB026840 (2023).
- Moussallam, Y. et al. Mantle plumes are oxidised. *Earth and Planetary Science Letters* **527**, 115798, doi:https://doi.org/10.1016/j.epsl.2019.115798 (2019).
- Obata, M., Ozawa, K., Naemura, K. & Miyake, A. Isochemical breakdown of garnet in orogenic garnet peridotite and its implication to reaction kinetics. *Mineralogy and Petrology* **107**, 881-895, doi:10.1007/s00710-012-0260-4 (2013).
- Pan, F.-B., Jin, C., He, X., Tao, L. & Jia, B.-J. A plate-mantle convection system in the west Pacific revealed by Tertiary ultramafic-mafic volcanic rocks in Southeast China. *Earth and Space Science* **8**, e2020EA001324, doi:https://doi.org/10.1029/2020EA001324 (2021).
- Pearson, D. et al. Hydrous mantle transition zone indicated by ringwoodite included within diamond. *Nature* **507**, 221-224 (2014).
- Rohrbach, A. et al. Metal saturation in the upper mantle. *Nature* **449**, 456-458, doi:10.1038/nature06183 (2007).
- Rohrbach, A. & Schmidt, M. W. Redox freezing and melting in the Earth's deep mantle resulting from carbon-iron redox coupling. *Nature* **472**, 209-212 (2011).
- Revenaugh, J. & Sipkin, S. Seismic evidence for silicate melt atop the 410-km mantle discontinuity. *Nature* **369**, 474-476 (1994).
- Stagno, V., Ojwang, D. O., McCammon, C. A. & Frost, D. J. The oxidation state of the mantle and the extraction of carbon from Earth's interior. *Nature* **493**, 84-88, doi:10.1038/nature11679 (2013).
- Špaček, P., Ackerman, L., Habler, G., Abart, R. & Ulrych, J. Garnet Breakdown, Symplectite Formation and Melting in Basanite-hosted Peridotite Xenoliths from Zinst (Bavaria, Bohemian Massif). *Journal of Petrology* **54**, 1691-1723, doi:10.1093/petrology/egt028 (2013).
- Tauzin, B., Debayle, E. & Wittlinger, G. Seismic evidence for a global low-velocity layer within

- the Earth' s upper mantle. *Nature Geoscience* **3**, 718-721, doi:10.1038/ngeo969 (2010).
- Takei, Y. Effects of Partial Melting on Seismic Velocity and Attenuation: A New Insight from Experiments. *Annual Review of Earth and Planetary Sciences* **45**, 447-470, doi:<https://doi.org/10.1146/annurev-earth-063016-015820> (2017).
- van Aken, P. A., Liebscher, B. & Styrsa, V. J. Quantitative determination of iron oxidation states in minerals using Fe L_{2,3}-edge electron energy-loss near-edge structure spectroscopy. *Physics and Chemistry of Minerals* **25**, 323-327, doi:10.1007/s002690050122 (1998).
- Wang, X.-C., Wilde, S. A., Li, Q.-L. & Yang, Y.-N. Continental flood basalts derived from the hydrous mantle transition zone. *Nature Communications* **6**, 7700 (2015).
- Wang, Z.-Z., Liu, S.-A., Chen, L.-H., Li, S.-G. & Zeng, G. Compositional transition in natural alkaline lavas through silica-undersaturated melt–lithosphere interaction. *Geology* **46**, 771-774, doi:10.1130/g45145.1 (2018).
- Wei, W., Xu, J. D., Zhao, D. P. & Shi, Y. L. East Asia mantle tomography: New insight into plate subduction and intraplate volcanism. *Journal of Asian Earth Sciences* **60**, 88-103, doi:10.1016/j.jseaes.2012.08.001 (2012).
- Xu, R. *et al.* Decoupled Zn-Sr-Nd isotopic composition of continental intraplate basalts caused by two-stage melting process. *Geochimica et Cosmochimica Acta* **326**, 234-252, doi:<https://doi.org/10.1016/j.gca.2022.03.014> (2022).
- Yang, J. & Faccenda, M. Intraplate volcanism originating from upwelling hydrous mantle transition zone. *Nature* **579**, 88-91 (2020).
- Zeng, G. *et al.* Nephelinites in eastern China originating from the mantle transition zone. *Chemical Geology* **576**, 120276, doi:<https://doi.org/10.1016/j.chemgeo.2021.120276> (2021).

Responses to the comments from the reviewers (in blue color)

Reviewer #1 (Remarks to the Author):

The authors have addressed all my concerns and provided comprehensive responses to my comments. The manuscript has improved significantly in terms of clarity and quality. However, I still believe that the first issue I raised (MTZ origin for the xenoliths) is worth further investigation. Providing additional evidence, such as other compositional information from MTZ sources, would strengthen the argument and enhance the manuscript's persuasiveness.

That said, the paper is internally consistent and well-structured. I recommend publication either in its current form or after minor revisions to further refine the discussion.

Response: Thank you for your constructive suggestions and recognition of our work. We will continue to study these interesting samples and aim to provide additional information in follow-up work.

Reviewer #2 Evaluations:

The revised version of “Iron disproportionation in peridotite fragments from the mantle transition zone” is more to the point and addresses the questions/aspects I raised in the review properly, either in the main text or in the rebuttal. Specifically, the authors use now the Beyer geobarometer to calculate pressures, provide an explanation for the decoupling of Na and Si in the garnet, clarify how eclogitic garnets are incorporated in lherzolitic xenoliths provide an explanation how Fe³⁺ is incorporated in olivine. Also, in my opinion, the concerns of reviewer 1 are adequately addressed but I'd leave that decision to the other reviewer or the editors, of course. Overall, I'd be happy to see this manuscript being published in Nature Communications, to me it is high quality work on an exciting topic. I have some additional minor comments the authors might adopt.

1. Maybe elaborate a bit on the uncertainties of the pressure calculation. Given the rather large variation for the Na measurements in garnet, the reader might be interested how this translates into a 2 sigma uncertainty of P.

Response: Thank you for your thoughtful comments. In response to your suggestion, we have added 2 sigma uncertainties for the major element compositions in revised supplementary Table 4. Additionally, and exhibited the 2 sigma uncertainties for the calculated pressures are now shown in the revised Figure 2a.

2. Looking again at the garnet compositions and on the criteria to distinguish eclogitic from peridotitic garnet (Fig. 7 of Beyer and Frost), it seems that garnets M1-M7 are rather (besides

the Na content) transitional in Cr content and Ca# and belong clearly to the peridotitic suite in terms of Mg# and Ti content. Maybe it would be interesting for the reader to address this apparent contradiction.

Response: We agree that the garnets show some transitional features in composition. As noted in the text (lines 146-149), these Na-rich majoritic garnets are likely reaction products of carbonated MORB within the MTZ, which could modify their chemical compositions. We have added this possible interpretation in the revised manuscript (lines 149-151).

3. Fig. 2 in the rebuttal letter is very nice, maybe consider to put it into the Supplement.

Response: We appreciate your valuable insights. Crystal defects in olivine (Ol₃) may play an important role in the storage of both Fe³⁺ and H⁺ (McCammon et al., 2004; Muir et al., 2023), as well as in the rheological softening of deep mantle rocks (Cline et al., 2018). We believe these important geological processes warrant a more in-depth discussion, which we plan to address in future work rather than in the current supplemental material.

4. Line 103 (file with track changes) lithophile

Response: We have corrected it as suggested.

5. Line 135 (file with track changes) I'd rather say "calculated" and give uncertainties

Response: We have corrected it as suggested.

Cited references

- Beyer, C. & Frost, D. J. The depth of sub-lithospheric diamond formation and the redistribution of carbon in the deep mantle. *Earth and Planetary Science Letters* **461**, 30-39, doi:<https://doi.org/10.1016/j.epsl.2016.12.017> (2017).
- Cline II, C., Faul, U., David, E., Berry, A. & Jackson, I. Redox-influenced seismic properties of upper-mantle olivine. *Nature* **555**, 355-358 (2018).
- McCammon, C. et al. Oxidation state of iron in hydrous mantle phases: implications for subduction and mantle oxygen fugacity. *Physics of the Earth and Planetary interiors* **143**, 157-169 (2004).
- Muir, J. M. R., Jollands, M. & Zhang, F. The oxidation states of Iron in dry and wet olivine: a thermodynamic model. *Journal of Geophysical Research: Solid Earth* **128**, e2023JB026840, doi:<https://doi.org/10.1029/2023JB026840> (2023).